Watching eyes on potential litter can reduce littering: evidence from two field experiments

Bateson Melissa 1
Robinson Rebecca 2
Abayomi-Cole Tim 2
Greenlees Josh 2
O’Connor Abby 2
Nettle Daniel 1 daniel.nettle@ncl.ac.uk
1 Centre for Behaviour and Evolution, Newcastle University , Newcastle , United Kingdom
2 School of Psychology, Newcastle University , Newcastle Tyne and Wear , United Kingdom
Deaner Robert
Electronic publication date: 2015 Dec 1
Publication date: 2015
Volume: 3
Electronic Location ID: e1443
Received 2015 Oct 8; Accepted 2015 Nov 4
Copyright: © 2015 Bateson et al.
Copyright year: 2015
Copyright holder: Bateson et al.
License: This is an open access article distributed under the terms of the Creative Commons Attribution License, which permits unrestricted use, distribution, reproduction and adaptation in any medium and for any purpose provided that it is properly attributed. For attribution, the original author(s), title, publication source (PeerJ) and either DOI or URL of the article must be cited.
License URL: https://creativecommons.org/licenses/by/4.0/

Keywords: Littering, Antisocial behaviour, Watching eyes, Natural surveillance, Interventions, Cooperation

Funding: The authors received no funding for this work.

==============================
Littering constitutes a major societal problem, and any simple intervention that reduces its prevalence would be widely beneficial. In previous research, we have found that displaying images of watching eyes in the environment makes people less likely to litter. Here, we investigate whether the watching eyes images can be transferred onto the potential items of litter themselves. In two field experiments on a university campus, we created an opportunity to litter by attaching leaflets that either did or did not feature an image of watching eyes to parked bicycles. In both experiments, the watching eyes leaflets were substantially less likely to be littered than control leaflets (odds ratios 0.22–0.32). We also found that people were less likely to litter when there other people in the immediate vicinity than when there were not (odds ratios 0.04–0.25) and, in one experiment but not the other, that eye leaflets only reduced littering when there no other people in the immediate vicinity. We suggest that designing cues of observation into packaging could be a simple but fruitful strategy for reducing littering.

General Introduction

Littering constitutes a major societal problem. Litter is perceived as unsightly and deleterious to quality of life. It can cause health and safety problems as well as contributing to environmental contamination. Moreover, there is evidence that the presence of litter in an environment can increase the prevalence of other social problems such as crime through what has been termed ‘the spreading of disorder’ (Keizer, Lindenberg & Steg, 2008). Furthermore, it is well established that ‘littering begets littering’ (Kraus, Freedman & Whitcup, 1978; Cialdini, Reno & Kallgren, 1990; Huffman et al., 1995; Keizer, Lindenberg & Steg, 2008; Schultz et al., 2011; Weaver, 2015). Therefore, any intervention that reduces littering behaviour has the potential to produce large and synergistic benefits.

Our recent research on littering is based on the ‘watching eyes effect.’ This is the finding that placing images of human eyes in participants’ environments often causes them to behave in a more prosocial manner than they otherwise would (Haley & Fessler, 2005; Bateson, Nettle & Roberts, 2006; Burnham & Hare, 2007; Keller & Pfattheicher, 2011; Oda et al., 2011; Nettle et al., 2012; Powell, Roberts & Nettle, 2012; Baillon, Selim & Van Dolder, 2013; Sparks & Barclay, 2013). Actions observed by others can have social and reputational consequences, whereas those that go unobserved cannot. Thus, individuals are highly sensitive to cues indicative of observation, and it seems that even subtle (and in the case of artificial eye images, false) cues are in some instances sufficient to modulate behaviour, producing behavioural decisions more likely to meet the approval of others.

Three previous studies have investigated whether the watching eyes effect can be used to reduce littering and encourage proper disposal (Ernest-Jones, Nettle & Bateson, 2011; Francey & Bergmuller, 2012; Bateson et al., 2013). These studies used wall-mounted images of eyes at locations where individuals had either naturally occurring or experimentally-provided opportunities to litter. All three studies found evidence that the eye images significantly reduced littering rates or improved proper disposal compared to control images or locations. In two of the studies (Ernest-Jones, Nettle & Bateson, 2011; Bateson et al., 2013), it was shown that the effect did not require the display of any explicit message concerning littering. Participants presumably know that littering is perceived negatively; this and the general connection between being watched and the desire for a positive social reputation (Oda et al., 2011) appear to be sufficient for the effect to work.

Whilst these results were encouraging, it is not practical to envisage displaying wall-mounted watching eyes signs in every location where littering could occur. A more promising strategy might be to transfer the watching eyes onto the potential litter itself, in the form of images on packaging. The principle of placing anti-littering interventions onto packaging is very well established. The limited experimental evidence suggests that such interventions can have positive effects, but do not always do so: Wever et al. (2010), for example, found that an obtrusive verbal message on cups reduced littering rates, but a less obtrusive icon and message did not. However, to date no study has examined the effect of placing images of watching eyes onto potential items of litter on littering rates.

In this paper, we present the results of two experiments designed to do this.

Several previous watching eyes studies have found interactions between the presence of real human observers in the vicinity and the effectiveness of the artificial images of eyes (Ernest-Jones, Nettle & Bateson, 2011; Powell, Roberts & Nettle, 2012; Ekström, 2012; Bateson et al., 2013). The usual form such interactions take is that the eye images are more effective (or only effective) when there are few or no real observers nearby (Ernest-Jones, Nettle & Bateson, 2011; Powell, Roberts & Nettle, 2012; Ekström, 2012). This makes sense: a real person is presumably a much stronger cue of social observation than an artificial image. It thus seems unlikely that images of eyes should have any incremental effect on behaviour if there are already real observers in the environment. Bateson et al. (2013) found a different pattern of interactions, namely that eye images were more effective when the environment was very crowded than when it was moderately crowded. The authors suggested that this might be reconciled with other findings by assuming a non-monotonic relationship between the number of people in a location and the social attention paid to any particular individual.

If a real person is a more potent cue of social observation than a mere image, the presence of real observers should do more than just moderate the effectiveness of artificial eye images. It should also be a significant predictor of behaviour in its own right. Applied to littering, we should expect that the proximity of other people would be a negative predictor of littering at least as strong as the presence of artificial eye images. Ernest-Jones, Nettle & Bateson (2011) found evidence consistent with this principle. People littered less when there were more people in proximity, with the sharpest contrast being between people who were alone and those who were in small groups with 1–3 others. This connects to the idea of ‘natural surveillance’ from the urban studies literature (Jacob, 1961): urban spaces generate the fewest social problems when they are busy and individuals are not hidden from one another. The impact of natural surveillance is not widely documented in the littering literature: the largest observational study of littering (Schultz et al., 2011) concluded that being in a social group or in proximity to others had no effect on littering probability. Thus, the effect of natural surveillance on littering behaviour requires clarification.

We therefore sought to examine the effects of artificial eye images and the proximity of real people on littering behaviour in a field setting. As in one of our previous studies (Bateson et al., 2013), we used the classic approach of generating an experimental opportunity to litter by presenting people with a leaflet they would be likely to wish to dispose of, and observing what they did with it (Cialdini, Reno & Kallgren, 1990; Keizer, Lindenberg & Steg, 2008). In our case, the leaflet was attached to people’s bicycles while they were parked. Whereas Bateson et al. (2013) manipulated the presence of wall-mounted signs with eye images, we manipulated whether or not there were eye images on the leaflet itself.

Experiment 1

Introduction

In experiment 1, the experimental conditions consisted of a simple contrast between a leaflet showing a prominent image of watching eyes (the eyes condition), and the same leaflet with the eyes part obscured (the control condition). The leaflet made no mention of littering, since our previous work suggested that explicit verbal messages about littering were not necessary for the watching eyes effect on littering to occur. We also recorded the number of people other than the observed participant in the immediate vicinity. We predicted that the probability of littering would be lower in the eyes than the control condition, and lower with other people in the vicinity than without. We also predicted an interaction between experimental condition and proximity of other people, such that the effect of experimental condition would be stronger when there were no real observers than when there were.

Methods

Ethics statement

Ethical approval was obtained from the Psychology subcommittee of the Faculty of Medical Sciences ethics committee, Newcastle University, approval number 278945. As participants were observed in a public place, and were not individually identified or approached during the course of the experiment, it was not required or appropriate to obtain informed consent or conduct debriefing.

Study site and participants

Data collection took place at two bicycle racks outside buildings on Newcastle University campus used by large numbers of students and staff. Neither of the racks featured the large wall-mounted watching eyes posters directed at cycle thieves that we have installed as part of previous research (Nettle, Nott & Bateson, 2012; Bateson et al., 2013). Both sites were artificially lit and had litter bins in the vicinity.

Experimental design

We generated a standardized opportunity to litter by making informational leaflets with a message unrelated to littering (“Beware of bike thieves. Lock your bike.”), printed in black and white on A5 paper. In the centre of the leaflet was an image of large pair of direct-gaze male eyes with the legend “Cycle thieves. We are watching you” and the logos of the local police and university (Fig. 1). This image has been used on large wall-mounted signs in anti-cycle theft campaigns at Newcastle University (Nettle, Nott & Bateson, 2012). The experimental leaflet was likely to have been credible, since the university security service does run leaflet campaigns to encourage students to lock their bicycles, and the watching eyes image we used on the leaflet carries official university branding and is a familiar sight around campus.

Figure 1 Leaflets used in the experiments.

Note that the leaflets were printed in black and white in experiment 1 and colour in experiment 2. (A) The control condition of both experiments. (B) The eyes condition of experiment 1/large eyes condition of experiment 2. (C) The small eyes condition used in experiment 2 only.

In the control condition, the eyes of the watching eyes image were completely obscured by an overlaid graphic of a bicycle lock. This was a relatively strong control, as the university and police logos and the words “Cycle thieves. We are watching you” were still clearly visible. Thus, any effect of the semantic activation of officialdom or the police, or verbal invocation of surveillance, would be equally present in the control and eyes conditions: the only difference was the presence of the eyes image itself. Leaflets were folded and attached to one end of the handle-bars of bicycles at the rack using an elastic band (Fig. 2). The position was intended to make it difficult for the cyclist to leave with their bicycle without first removing the leaflet.

Figure 2 Illustration of the position of a leaflet on bicycle handlebars.

Data recording

The behavioural decisions of cyclists returning to their bicycles were unobtrusively recorded by an observer (Rebecca Robinson) situated at least 20 m away. As in our previous study (Bateson et al., 2013), the categories of behavioural decision recorded were: left without removing the leaflet from its position; kept it on the person (e.g., in a pocket or bag); placed it in the rubbish bin; put elsewhere in the vicinity (for example onto another bicycle); or threw it onto the ground. In addition to the behavioural decision, we recorded the participant’s approximate age, sex, and the number of other people in the immediate vicinity (a radius of approximately 6 m) at the time the interaction with the leaflet occurred. Data were obtained in 7 sessions of recording between 30th January and 28th February 2014 on weekdays during university teaching hours. All occurred in daylight, with either fine weather or light rain. Only one condition (eyes or control) was run during a session. Both conditions were run in both locations.

Data analysis

For the purposes of statistical analysis, we assumed that each returning cyclist constituted an independent unit of analysis. This assumption is reasonable in view of the large numbers of people that use both of the study sites, the observation sessions being held at differing times on different weekdays, and the asynchronous arrival of cyclists during the sessions. We required a binary dependent variable of littering or no littering. We classified throwing the leaflet on the ground as littering, and keeping the leaflet on person, placing it in a rubbish bin, or placing it elsewhere in the environment as not littering. Cases where the participant left without removing the leaflet from its position were excluded, since we could not be sure that the participant had noticed the leaflet and hence could not justifiably treat them as having decided not to litter. This classification scheme was identical to that used in our previous study (Bateson et al., 2013), and its use was preplanned. However, leaving the leaflet elsewhere in the environment (which was almost always on another bicycle) is ambiguous. It is not the overtly antisocial behaviour our study was seeking to reduce, and it could be that the participant wants another cyclist to have the information on the leaflet. However, it is a quick way of getting rid of the leaflet. We have therefore repeated all the analyses with cases of leaving the leaflet elsewhere in the environment excluded rather than treated as not littering, and the results are substantively the same.

There was limited variation in age (98% of participants judged to be under 25) and sex (86% of participants judged to be male), and so effects of age and sex were not considered further. The number of other people in the vicinity had a highly right-skewed distribution with a mode of 0 (range 0–6). We therefore dichotomized it into whether there was someone else in the vicinity or not for the purposes of the analysis.

As our dependent variable was dichotomous, we modeled the data using a generalized linear model with binomial error structure and a logit link function, using the ‘glm’ function in R (R Core Development Team, 2013). The logit link is the default link function for a binomial error structure, and has the advantage that its coefficients are interpretable as log odds ratios. The independent variables were condition (eyes vs. control), someone else in vicinity (true/false), and the condition by someone in vicinity interaction. The model was repeated using a generalized linear mixed model with an additional random effect of site, but the differences made by the addition of the random effect were trivial and so the results of this model are not shown.

Results

The raw data from experiment 1 are downloadable as Supplemental Information 1. There were a total of 316 observations. The breakdown of behavioural decisions observed is shown in Table 1. Dichotomization of these into ‘littered’ and ‘did not litter’ produced 31 instances of littering and 253 of not littering. Figure 3A shows the proportion of participants littering by condition and whether there was someone else in the vicinity. The figure suggests a greater probability of littering when there was no-one else in the vicinity than when there was someone else, and in the control condition compared to the eyes condition, regardless of whether there was someone else in the vicinity or not. This was borne out by the statistical model. The model fit the data substantially better than a null model (likelihood ratio test, χ32=25.17, p < 0.01). There were significant main effects of condition (B = − 1.13, s.e.(B) = 0.55, z = − 2.05, p = 0.04) and presence of someone else in the vicinity (B = − 1.38, s.e.(B) = 0.47, z = − 2.92, p < 0.01), but the interaction between condition and presence of someone else was not significant (B = − 0.62, s.e.(B) = 0.96, z = − 0.65, p = 0.52).

Figure 3 Probability of littering by experimental condition and presence of someone else in the vicinity.

(A) Experiment 1. (B) Experiment 2.

Table 1 Summary of behavioural decisions (number and % of observations within condition) observed in experiment 1.

Behavioural decision	Overall	Control	Eyes	Classification	
Left without removing leaflet	32 (10.1%)	10 (6.8%)	22 (13.0%)	Excluded	
Kept on person	188 (59.5%)	84 (57.1%)	104 (61.5%)	Did not litter	
Put in bin	0 (0%)	0 (0%)	0 (0%)	Did not litter	
Put elsewhere	65 (20.6%)	30 (20.4%)	35 (20.7%)	Did not litter	
Threw on ground	31 (9.8%)	23 (15.6%)	8 (4.7%)	Littered	

Discussion

In line with our predictions, there were significant negative effects of the watching eyes image and of there being other people in the vicinity on littering. Coefficients from logistic regression can be interpreted as log odds ratios, and so the eyes condition coefficient of −1.13 corresponds to an odds ratio of 0.32 for people to litter when the leaflet showed eyes than when it did not. The ‘someone else in the vicinity’ coefficient of −1.38 corresponds to an odds ratio of 0.25 for littering when someone else is in the vicinity compared to when no-one was. These would be considered large effects (Cohen, 1988).

Contrary to our prediction, there was no evidence of an interaction between the presence of other people in the vicinity and the experimental condition. The interaction we predicted, based on previous findings (Ernest-Jones, Nettle & Bateson, 2011; Powell, Roberts & Nettle, 2012; Ekström, 2012), was that the effects of watching eyes image on the leaflet would be larger when no real people were in the vicinity. In fact, the effect appeared uniform whether there were others in the vicinity or not (see Fig. 3A). We were not able to establish whether the different interaction effect found by Bateson et al. (2013), with the watching eyes manipulation having a larger effect when the area was very crowded than when it was moderately crowded, was present. That interaction was driven by a large eyes effect when the number of other people in the vicinity exceeded six. There were many such instances in the previous study, but none at all in the present one. This may reflect differences in time and day of recording, and chance coincidence with busy university events.

The prevalence of littering we observed was substantially lower (10.9%) than in our previous study using the same method (Bateson et al., 2013; 23.2% in the no-litter conditions ). This was partly due to the large experimental effect we observed in the present study; the two most directly comparable non-eyes conditions, the control condition of the present experiment and the no eyes, no litter condition of Bateson et al. (2013), produced fairly similar littering rates (16.8% and 20.1% respectively).

Experiment 2

Introduction

The results of experiment 1 suggested that placing an image of watching eyes on a piece of potential litter did influence people’s littering decisions, making them less likely to do so. We felt it important to replicate this result. Moreover, the watching eyes image on the experimental leaflet was very large, occupying a substantial portion of the leaflet’s area. It would not be feasible to persuade manufacturers to devote such a large portion of their packaging to an anti-littering intervention. In experiment 2, as a first step towards a more translatable intervention, we experimented with reducing the size of the watching eyes image, to examine the impact on its effectiveness. Experiment 2 therefore repeated the procedure of experiment 1, but adding a third condition in which the watching eyes stimulus was reduced in size. In this ‘small eyes’ condition, both the width and height of the stimulus were reduced to one third of their previous values, meaning that the total area was reduced by around 89% (see Fig. 1). As before, we predicted that there would be a main effect of eyes (lower probability of littering in either eyes condition compared to control), a main effect of people in the vicinity (lower probability of littering when other people were in the vicinity), and an interaction between condition and other people in the vicinity (eyes conditions having a greater effect when there were few other people in the vicinity). We were particularly interested in whether any watching eyes effect would be weaker for the small eyes leaflet compared the large eyes.

Methods

Ethics statement

Ethical approval was obtained from the Psychology subcommittee of the Faculty of Medical Sciences ethics committee, Newcastle University, approval number 753137, on the same basis as for experiment 1.

Study sites, participants and data recording

Data collection was carried out on rain-free days between December 1st 2014 and January 30th 2015 using the same protocol and coding scheme as experiment 1, although in experiment 2 the leaflets were reproduced in colour rather than black and white. A third site within the campus was added to the two used in experiment 1. Data were collected by either Tim Abayomi-Cole, Josh Greenless or Abby O’Connor. Pilot sessions were conducted with all three researchers present to establish uniform scoring criteria. The control and small eyes conditions were run at all three sites, and the large eyes condition at two of the three (the final site/condition combination was omitted in error). Due to the short day length during December and early January, the sun had set before the end of one session in each condition 8% of the total number of observations were from after sunset. However, all study sites were artificially lit.

Data analysis

As in experiment 1, there was limited variation in participant age (92% judged to be under 25) and sex (77% male), and so age and sex are not considered further. Number of other people in the vicinity again showed a right-skewed distribution with a mode of 0 (range 0–6) and was therefore dichotomized into someone else in vicinity versus no one else in vicinity, as in experiment 1. The statistical model was the same as that for experiment 1 save for the extra coefficients due to the extra experimental condition. As in the previous study, repeating the model with a random effect of site produced only trivial differences and the results of that model are not shown. To compare the results with those of experiment 1, we created a meta-analytic plot of the main effects of both studies using the ‘metafor’ package.

Results

The raw data from experiment 2 are downloadable as Supplemental Information 2. We made a total of 396 observations. The breakdown by behavioural decision is shown in Table 2. Dichotomization of the behavioural decision produced 56 instances of littering and 257 of not littering. Figure 3B shows the probability of littering by experimental condition and presence of someone else in the vicinity. The figure suggests that both eyes conditions substantially reduced the probability of littering compared to the control condition, but only when there was no-one else in the vicinity. The statistical model confirms this. The model fit the data substantially better than a null model (likelihood ratio test, χ52=32.12, p < 0.01). There was a significant main effect of presence of someone else in the vicinity (B = − 3.12, s.e.(B) = 1.05, z = − 2.98, p < 0.01). The effect of the large eyes condition relative to control was significant (B = − 1.07, s.e.(B) = 0.47, z = − 2.28, p = 0.02), as was the effect of the small eyes condition relative to control (B = − 1.51, s.e.(B) = 0.55, z = − 2.78, p < 0.01). There were also significant interactions between the large eye condition and someone else in the vicinity (B = 3.40, s.e.(B) = 1.17, z = 2.92, p < 0.01) and the small eye condition and someone else in the vicinity (B = 2.57, s.e.(B) = 1.23, z = 2.10, p < 0.01). These interactions were driven by the fact that the probability of littering was lower for both eyes conditions than control when no-one else was in the vicinity (coefficients of -1.07 for large eyes and −1.51 for small eyes), but actually higher than the control conditions when there was someone else in the vicinity (coefficients of 2.33 for large eyes and 1.06 for small eyes).

Table 2 Summary of behavioural decisions (number and % of observations within condition) observed in experiment 2.

Behavioural decision	Overall	Control	Large eyes	Small eyes	Classification	
Left without removing leaflet	83 (21.0%)	53 (35.3%)	11 (10.1%)	19 (13.8%)	Excluded	
Kept on person	205 (51.8%)	63 (42.0%)	58 (53.7%)	84 (60.9%)	Did not litter	
Put in bin	11 (2.8%)	0 (0%)	5 (4.6%)	6 (4.3%)	Did not litter	
Put elsewhere	41 (10.4%)	9 (6.0%)	14 (13.0%)	18 (13.0%)	Did not litter	
Threw on ground	56 (14.1%)	25 (16.7%)	20 (18.5%)	11 (8.0%)	Littered	

Figure 4 shows a meta-analytic plot of the main effects of experiments 1 and 2 for both the experimental conditions, and the impact of there being someone else in the vicinity. As the figure shows, the effect sizes from the two watching eyes conditions in experiment 2 are similar to one another, and similar to the single effect size from experiment 1: odds ratios of the order of 0.3 for littering when the leaflet has watching eyes compared to control. Figure 4 also shows the effect sizes for the presence of someone else in the vicinity. These effects were of similar magnitude to those of the watching eyes manipulation, and their confidence intervals for experiments 1 and 2 overlapped.

Figure 4 Meta-analytic forest plot of the main effects observed in experiments 1 and 2.

Parameter estimates from the statistical models and their standard errors have been transformed into odds ratios and their 95% confidence intervals.

Discussion

The pattern of behavioural decisions observed was similar to that seen in experiment 1, and in our previous study (Bateson et al., 2013), with most participants keeping the leaflet on their person, and a minority throwing them on the ground. The overall rate of littering in experiment 2 (17.9%) was intermediate between that of the present experiment 1 and that of Bateson et al. (2013). Experiment 2 replicated the main effects of experiment 1: littering was significantly less frequent when there was someone else in the vicinity than when there was no-one, and significantly less frequent in the eyes than the control conditions. The sizes of these effects were of similar magnitude as those observed in experiment 1 (see Fig. 4). There was no indication at all that making the watching eyes image much smaller reduced its effectiveness: the effect size for small eyes condition compared to control was comparable to that for large eyes versus control.

Whilst the main effects were similar between experiments 1 and 2, the interaction effects were not. In experiment 2, as predicted and in contrast to experiment 1, the watching eyes manipulation was effective in reducing littering only when there were no real observers in the vicinity. This was in keeping with previous research (Ernest-Jones, Nettle & Bateson, 2011; Powell, Roberts & Nettle, 2012; Ekström, 2012). However, the nature of the interaction effect was subtly different from those studies. For Ernest-Jones, Nettle & Bateson (2011) and Powell, Roberts & Nettle (2012), the watching eyes effect was attenuated when there were many real observers in the vicinity, but not reversed. Here, we observed a stronger crossover interaction: littering was actually more likely in the eyes than control conditions when there were observers in the vicinity (see Fig. 3B). This crossover was not predicted and the reasons for it are not clear. However, it was not sufficient to abolish the (negative) main effects of eyes on littering: littering was still less likely to occur overall in the eyes than control conditions. We were again unable to test for the pattern found in Bateson et al. (2013) where the watching eyes effect became stronger in very crowded locations, since we had no instances of more than six people in the vicinity.

General Discussion

In two separate experiments, we found that that the presence on a leaflet of an image of watching eyes substantially reduced the probability that the leaflet would be littered. The effects were large, with the eyes image reducing the odds of littering by around two thirds compared to a closely-matched control leaflet given out in the same locations. These results are consistent with body of other findings using various paradigms to show that displaying images of watching eyes can often increase prosocial behaviour (Haley & Fessler, 2005; Bateson, Nettle & Roberts, 2006; Burnham & Hare, 2007; Keller & Pfattheicher, 2011; Oda et al., 2011; Nettle et al., 2012; Powell, Roberts & Nettle, 2012; Baillon, Selim & Van Dolder, 2013; Sparks & Barclay, 2013), and with previous studies on watching eyes and littering more specifically (Ernest-Jones, Nettle & Bateson, 2011; Francey & Bergmuller, 2012; Bateson et al., 2013). One of the experiments, though not the other, confirmed the pattern seen in several previous studies where the effectiveness of the eye images was moderated by whether or not there were any real people in the vicinity (Ernest-Jones, Nettle & Bateson, 2011; Powell, Roberts & Nettle, 2012; Ekström, 2012).

In both experiments, we found a main effect of the presence of real observers in the vicinity. People were much less likely to litter when there were other people around them, concurring with one of our previous studies (Ernest-Jones, Nettle & Bateson, 2011). This effect was at least as strong as the experimental watching eyes image effect. In a sense this is a reassuring finding: it would after all be very strange if people’s behaviour responded to artificial cues of social observation, but was impervious to real cues. It highlights the important of ‘natural surveillance’—people using public spaces and not being hidden from one another—in the self-regulation of urban spaces (Jacob, 1961). A previous large observational study of littering (Schultz et al., 2011) did not detect any influence of the number of people in the vicinity. Had there been effects of anything like the size found in our experiments, that study would have had the statistical power to detect them. Thus, this discrepancy requires further investigation.

The experiments presented here have a number of scientific and practical limitations. On the scientific side, our design does not allow us to discriminate between, on the one hand, only eye images reducing littering and, on the other, any image invoking people being equally effective. Such controls have been investigated elsewhere, and the results suggest a specific effect of eyes rather than just the invocation of people (Baillon, Selim & Van Dolder, 2013). We also investigated only a single watching eyes image: a stern pair of male eyes we have used in previous research on the same campus, and hence whose use would not arouse suspicion. We have not shown whether the same effect could be achieved using other, less intimidating eye stimuli. However, we note that some watching eyes studies on behaviours other than littering have used stimuli including friendlier and more positive eye images, and still found significant eyes effects (Nettle et al., 2012; Powell, Roberts & Nettle, 2012; e.g., Baillon, Selim & Van Dolder, 2013). The results of the two experiments also differed somewhat. Since the protocol was varied in a number of minor ways beyond the addition of the extra condition (for example, printing in colour rather than black-and-white in experiment 2), the differences cannot be definitively explained without further experimentation. Finally, although our field experimental approach minimizes demand characteristics and produces data with high ecological validity, it does not provide any insight into the psychological processes underlying the observed effects. Other kinds of studies are required to shed light on these issues (see e.g., Oda et al., 2011; Pfattheicher & Keller, 2015).

The eye images we used, as well as being intimidating, were very large and prominent, much larger and more prominent than it would be feasible to persuade manufacturers to include on packaging. When we reduced their area in experiment 2, its effectiveness was completely undiminished. However, even the small eyes image remained very central to the leaflet (especially when folded on the bicycle handlebars), and the leaflet had little else in terms of visual elements. For practical usability, it would be important to establish that the effect could still be found using an eye image that was even smaller and less threatening in a design crowded with other visual elements. Perceptual research shows that directly-gazing faces capture processing priority in crowded visual scenes (Rothkirch et al., 2015). Furthermore, the stimuli used in some previous watching eyes studies have been extremely subtle (Haley & Fessler, 2005; Rigdon et al., 2009; Powell, Roberts & Nettle, 2012). Thus, it is plausible that a watching eyes image on packaging could be effective whilst being small and unobtrusive enough to be acceptable to manufacturers. However, further experimentation is required to establish unambiguously that this is the case.

Both the results presented here and those of our previous studies (Ernest-Jones, Nettle & Bateson, 2011; Bateson et al., 2013) support the view that it is not necessary to accompany the eye images with explicit verbal messages about littering. People know that littering is antisocial, and there appears to be a fundamental connection between observability and reluctance to be antisocial (Kraft-Todd et al., 2015). This is potentially very important: incorporating an anti-littering design element in packaging might be much more widely acceptable to manufacturers if it did not have to be obvious that this was its function. If it could simply be the face of someone endorsing or enjoying the product, then it is something that manufacturers might already incorporate anyway, or might need little persuasion to do so. We have not, however, tested the effectiveness of our watching eyes type intervention against a package element explicitly discouraging littering or encouraging proper disposal. Explicit messages have been shown to be effective in reducing littering in general (Huffman et al., 1995), and there is a limited amount of experimental evidence that they can be effective when placed on the potential litter items (Wever et al., 2010). Direct assessment of the relative effectiveness of watching eyes images and explicit litter-related messages on packaging, and the potential for combining these two components to maximize effectiveness, would be useful avenues for future research.

In view of litter’s negative aesthetic and environmental impacts, the tendency for litter to beget more litter, and the tendency of litter to promote other forms of disorder, the potential societal benefits from cheap interventions that reduce littering by even a small amount are very large. We have presented initial evidence here that placing images of watching eyes onto packaging could reduce littering and promote correct disposal. We hope that designers, manufacturers and regulators might be able to develop, refine and implement this principle in the future.

Supplemental Information

Supplemental Information 1 Data from experiment one

Click here for additional data file.

Supplemental Information 2 Experiment two data

Click here for additional data file.

Supplemental Information 3 R script to reproduce data analysis

Click here for additional data file.

Additional Information and Declarations

Competing Interests

Author Contributions

Human Ethics

Data Availability

The authors declare there are no competing interests.

Melissa Bateson conceived and designed the experiments, analyzed the data, contributed reagents/materials/analysis tools, reviewed drafts of the paper.

Rebecca Robinson, Tim Abayomi-Cole, Josh Greenlees and Abby O’Connor conceived and designed the experiments, performed the experiments, analyzed the data, contributed reagents/materials/analysis tools, prepared figures and/or tables, reviewed drafts of the paper.

Daniel Nettle conceived and designed the experiments, analyzed the data, contributed reagents/materials/analysis tools, wrote the paper, prepared figures and/or tables.

The following information was supplied relating to ethical approvals (i.e., approving body and any reference numbers):

Faculty of Medical Sciences ethics committee, Newcastle University

Approval numbers 278945 and 753137.

The following information was supplied regarding data availability:

We have provided all of the raw data plus the R script for data analysis as Supplementary Files.

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
