# Peer review of "Watching eyes on potential litter can reduce littering: evidence from two field experiments"

_PeerJ, doi:10.7717/peerj.1443_

## Round 0.1 · original submission · Minor Revisions

Dear Dr. Bateson and colleagues,

I have received reviews from three experts regarding your submission.

All three reviewers were supportive of the manuscript, but each raised some substantial concerns. I have read the manuscript and I find that most of these concerns are valid and require careful consideration

Therefore, my decision is Minor Revision.

Reviewer 1 ·

Basic reporting

The paper is well-written, with regard to both language and structure. The two studies presented form a distinct step in a larger research line, and therefore merit publication by themselves. The literature referencing and discussion mainly place the work in the research line of the authors. This could and should be a bit wider.

Experimental design

The research design appears to be sound, although there are some aspects that need to be described more explicitly in the manuscript.
- A mention is made on visibility of leaflets ‘even after dark’. Hence, more details should be provided with regards to timing and weather conditions during the research time slots, to argue that this is unlikely to have influenced results.
- ‘litter begets litter’. Hence, it needs to be clearer what happened during the time slots of the experiment. Did more litter events occur during the end of the time slot, when there was already litter around? Or was litter cleared in between?
- The classification of ‘placed somewhere else in the environment’ as not littering is problematic. For one, this could lead to people finding two leaflets on their bike, but it is unclear how this was handled. Second, and more importantly, it could be argued that placing elsewhere in the environment is littering. Hence, I’d say you would have to add a statistical analysis of your data (not replace the existing one!) where you present the conclusions if both the ground and elsewhere in the environment would constitute littering. (I would consider this my major comment on the manuscript, and I would like to see if your conclusions hold in this alternative classification).
- The experiments differ in more ways than needed, with black-and-white versus color printing, period of the year (although they were quite close). Part of the second experiment wasn’t executed as planned. Possibly some of the issues identified above played a role. I would expect an explicit discussion of such factors in a limitations section.

Validity of the findings

- The data presentation needs to be improved. Table 1 and 2 now only present the total data categorized over behavior types. This should be split in multiple columns showing the data related to the two and three conditions respectively.

Additional comments

- Is there anything to report about the littering of the elastic bands that were used to attach the flyers to the bicycles?
- In future research I would expect an exploration of the expression in the depicted eyes. The eyes used now could be described as stern or watchful. To be acceptable to brand owners to put on their packaging, it would be helpful if more friendly eyes (e.g. related to a product endorsement as mentioned in the manuscript) would also be effective.
- I would mention earlier in the manuscript that leaflets were attached to the bikes, as the reader now is confused by the assumption that there was a person handing them out, which would mean there is always at least one person in the vicinity.
- The statement that ‘the majority of litter is deliberately dropped’ is rather strong to be supported by only a single reference. In a large littering project in Australia, multiple behaviors causing littering were identified. Not all would qualify as deliberate. [Curnow, Streker, Williams. (1997) Understanding Littering Behaviour in Australia: a Review of the Literature. Beverage Industry Environment Council: Pyrmont, Australia]
- I would have expected you to reference the work of Kraus et al. as earlier work that studied the effect of leaflet design on littering. [Krauss, Freedman, Whitcup. (1996) Field and laboratory studies of littering. In: The Psychology of Vandalism, Goldstein AP (ed.). Plenum Press: New York.]

·

Basic reporting

No Comments

Experimental design

No Comments

Validity of the findings

Were the results caused by the eyes effect? I mean that the eyes might not be necessary for the reduction of littering. Eyes are parts of human face and people have some animistic feeling. There is a possibility that the subjects hesitated to throw the leaflets on the ground because parts of a living thing were printed. On the other hand, they might be not reluctant to litter the leaflets a bicycle lock was printed because it was a mere goods. That is, you might get the same effect with a print of human body without eyes or of animal pictures. Please consider the possibility.

Another issue is the result that littering was more likely in the eyes condition than control condition when there were observers in the vicinity in the experiment 2, which contradicted to the results in the experiment 1. The authors only said that the reason were not clear and did not discuss further. However, if the eyes stimulus has its effect only when there is no one in the vicinity, and increases littering when there were observers, the eyes effect is useless for prevention of littering in real life. The authors should discuss in detail the reason why the effect reversed between the two experiments. For example, is there any difference in condition other than the existence of observers between the two experiments?

Reviewer 3 ·

Basic reporting

--

Experimental design

--

Validity of the findings

--

Additional comments

The paper examines the effect of watching eyes that were printed on informational leaflets on littering behavior (i.e., whether people discard these leaflets). It was really a pleasure reading this well-written article in which every paragraph had an effective reading flow and contains relevant information. Additionally, I appreciate that the authors dealt with a relevant **applied** research question. I have only minor concerns that can surely be addressed in a revision.
- What the authors should really reconsider is whether their intervention seems practical and realisable. I am not sure whether companies would print angry eyes on their products. Why should they?
- In case of the leaflets, some participants surely had the impulse to throw these away. The eyes reduced the occurrence of littering. But consider a product on which the eyes are presented in an even more subtle way and might not even (consciously or unconsciously) be noticed by the consumers because so much additional information is printed on the product. Would the authors still assume that watching eyes are a reliable method to reduce littering? Is it really “practical”, as the authors suggest (l.66)? I’m not sure.
- 25. In my perspective, “fight against litter” sounds somewhat too militant.
- 38. The sentence is not logical
- 48. Citation is needed.
- 103. What about the bystander effect?
- Please avoid disclosure of observers’ names (e.g., Rebecca Robinson)
- Rebecca Robinson was “situated some meters away”. How then is there no social presence? Maybe the observer already induced social presence and an even stronger effect of the eyes was inhibited. From this perspective it seems odd that the number of other people in the vicinity started with 0.
- One methodological problem: Data were obtained in seven sessions and the eyes were presented as I assume in three or four sessions (please provide information). There could be a problem with interfering third variables; for instance, if on control group days it was raining, participants might have been more likely to throw away the leaflets, not because of not being watched, but because of the bad weather (perhaps wanting to get home faster or due to being ego-depleted or annoyed). How did the authors deal with controlling for third variables? I suggest running each condition simultaneously at one place or changing the conditions several times a day to avoid this methodological problem.
- 201-208. Please provide information for readers unfamiliar with the statistical models (consider supplementary material if the insertion of this would interrupt reading flow). Why is a binomial error structure assumed? Why is a logit link function applied?
- Why is “site” modelled as a random effect? I would have modelled “site” as a fixed effect.
- How are the main effects coded? Plus/minus 1? If they are null-zero coded, the authors must not interpret the main effects.
- 235. Citation is needed.
- 262. “We felt …” Why?
- Study 2: I think that data collection during a winter in Newcastle, UK, was a hard time. Nonetheless, there were still some people riding their bike. Great!
- 323. Please provide simple slopes.
- 386. “very much”? Does this hold true for the small effect found in study 2?
- l.428 What about reactance?
- I suggest discussing the application of **angry** eyes. Do the authors expect effects when neutral eyes are used?
- I would really appreciate if the authors inspired future research by speculating about the psychological process going on in their research (and/or if the author preferred moderators that would also stimulate and inspire future research). What psychological process drive the watching eyes effect, adherence to social norms as discussed by Bateson et al. (2013, PLOS ONE)? Is it public self-awareness (Pfattheicher & Keller, 2015, European Journal of Social Psychology), or reputational concerns (Ekström, 2012, Experimental Economics; Pfattheicher, in press, Motivation and Emotion)? Social desirability? Do people avoid negative evaluations or emotions by others (as I assume is true when angry eyes are presented)? Or is it a cascade like watching eyes => one feels being watched => public self-awareness/reputational concerns => social desirable behavior/ adherence to social norms?

---

## Round 0.2 · accepted · Accept

You've done a superb job addressing the reviewers' concerns and made my job very easy. Thank you!